# Computational Study of Natural Compounds for the Clearance of Amyloid-Βeta: A Potential Therapeutic Management Strategy for Alzheimer’s Disease

**DOI:** 10.3390/molecules24183233

**Published:** 2019-09-05

**Authors:** Syed Sayeed Ahmad, Haroon Khan, Syed Mohd. Danish Rizvi, Siddique Akber Ansari, Riaz Ullah, Luca Rastrelli, Hafiz Majid Mahmood, Mohd. Haris Siddiqui

**Affiliations:** 1Department of Bioengineering, Faculty of Engineering, Integral University, Lucknow 226026, India; 2Department of Pharmacy, Abdul Wali Khan University Mardan 23200, Pakistan; 3Department of Pharmaceutics, College of Pharmacy, University of Hail, PO Box 2440, Ha’il – 81451, Saudi Arabia; 4Department of Pharmaceutical Chemistry, College of Pharmacy, King Saud University, P.O. Box: 2457, Riyadh 11451, Saudi Arabia; 5Medicinal, Aromatic and Poisonous Plants Research Center (MAPRC), College of Pharmacy, King Saud University, PO Box 2457, Riyadh 11451, Saudi Arabia; 6Dipartimento di Farmacia, University of Salerno, 84084 Fisciano, Italy; 7Department of Pharmacology, College of Pharmacy, King Saud University PO Box 2457, Riyadh 11451, Saudi Arabia

**Keywords:** Alzheimer’s disease, natural compounds, binding energy, docking, Z-dock

## Abstract

Alzheimer’s disease (AD) is a widespread dynamic neurodegenerative malady. Its etiology is still not clear. One of the foremost pathological features is the extracellular deposits of Amyloid-beta (Aβ) peptides in senile plaques. The interaction of Aβ and the receptor for advanced glycation end products at the blood-brain barrier is also observed in AD, which not only causes the neurovascular anxiety and articulation of proinflammatory cytokines, but also directs reduction of cerebral bloodstream by upgrading the emission of endothelin-1 to induce vasoconstriction. In this process, RAGE is deemed responsible for the influx of Aβ into the brain through BBB. In the current study, we predicted the interaction potential of the natural compounds vincamine, ajmalicine and emetine with the Aβ peptide concerned in the treatment of AD against the standard control, curcumin, to validate the Aβ peptide–compounds results. Protein-protein interaction studies have also been carried out to see their potential to inhibit the binding process of Aβ and RAGE. Moreover, the current study verifies that ligands are more capable inhibitors of a selected target compared to positive control with reference to ΔG values. The inhibition of Aβ and its interaction with RAGE may be valuable in proposing the next round of lead compounds for effective Alzheimer’s disease treatment.

## 1. Introduction

Alzheimer’s disease (AD) is a common progressive neurodegenerative disease affecting more than 46 million people worldwide. Figures in the USA for the year 2015 reported AD to affect over 5.3 million people, whereby in 2050, it is expected to grow to nearly 1 million new AD cases per year, with an estimated frequency to range from 11 million to 16 million [1]. Unfortunately, the etiology of AD is still not clear. One of the major pathological characteristics of AD tends to be extracellular deposition of Amyloid-beta (Aβ) peptides in senile plaques. The Aβ cascade-inflammatory hypothesis has been found to be the most probable therapeutic checkpoint for the treatment of AD [2]. Concerning structural perspectives, amyloid oligomers are spherical [3], surface-active entities [4] prone to form pore-like assemblies in the plasma membrane of brain cells [5].The circulating Aβ toxins are transported by the receptor for advanced glycation end products (RAGE), a multiligand receptor running transversely through the blood-brain barrier (BBB) into the brain. A RAGE-Aβ toxin interaction at the BBB directs the induction of oxidative stress, inflammatory responses and decreases the cerebral blood flow. RAGE specifically binds to the β-sheet fibrillar domain of Aβ toxins. Thereby, the regulation of RAGE action at the BBB, owing to its significance in disease progression, has been seen as a valuable treatment strategy for AD patients [6]. The deposition of amyloid in tissues changes its normal function, and it has been noticed that its elevated concentrations exert nonspecific toxic effects on cells by disturbing the integrity of membranes. Similarly, deposition of amyloid in tissues has been directly correlated with enhanced appearance of RAGE. In the brain of AD patients, the appearance of RAGE increases in glia and neurons. The penalty of Aβ ligation of RAGE has been shown to be relatively dissimilar for neurons as compared to microglia, where microglia become activated as a result of Aβ-RAGE interaction, as reflected by amplified motility and the expression of cytokines [7,8,9,10,11]. 

Curcumin, the very active constituent of turmeric, has various valuable properties, including anti-inflammatory, antioxidant and antitumor effects. Different studies so far have suggested that curcumin decreases the level of amyloid. Oxidized proteins avoid memory deficits and are thus helpful to patients with AD [12]. Inhibitors of Aβ aggregation can work by providing an adverse surface in amyloid oligomeric interaction [13]. Curcumin typically binds the target protein/peptide in the amyloidogenic pathway which makes it effiecient as an anti-aggregation agent [14,15]. This fact has been confirmed by staining studies of the amyloid deposits in in vivo systems [16,17]. Curcumin acts as an efficient β-sheet breaker in interactions with the Aβ peptide [18,19]. Following the above hypothesis in the present study, we have analyzed the Aβ aggregation inhibition potential of ajmalicine, emetine and vincamine compounds against curcumin as a standard control for their anti-Alzheimer’s potential. 

## 2. Results and Discussion

There is corpus evidence of AD being linked with oligomerization of β-Amyloid peptides [20]. Thereby, one of the procedures to adapt to AD is to discover compounds that can promote Aβ anti-aggregation and clearance [21]. The natural products or extracts reported in various preclinical and certain clinical studies provide valuable input to AD therapy [22,23]. Among them, curcumin is shown to inhibit Aβ aggregation and act as an antidote to Aβ-induced toxicity [14]. In the present investigation, Aβ was found to associate with Vnc through the amino acids Phe19, Phe20, Ala21, Asp23, Ile32, Gly33, Leu34, Met35 and Val36 (Figure 1C, Table 1); with Ajm through the amino acid residues Phe19, Phe20, Ala21, Gly22, Asp23, Ile32, Gly33,Leu34, Met35,and Val36 (Figure 1A, Table 1); and with Eme through the amino acid residues Ala21, Gl22, Asp23, Gly33, Leu34, Met35 and Val36 (Figure 1B, Table 1); as compared to standard curcumin, which was found to associate through the amino acid residues Ala21, Glu22, Asp23, Val24, Gly25, Leu34, Met35, Val36 and Gly37 (Figure 1D, Table 1). The amino acid residues Ala21, Asp23, Met35, and Val36 were seen in common interaction with selected ligand as well as curcumin in Aβ.

The free energy of binding for the interaction complexes ‘Vnc-Aβ’, ‘Ajm-Aβ’ and ‘Eme-Aβ’ were found to be −5.45, −6.66 and −6.99 kcal/mol respectively along with with their evaluated inhibition constants 249.96, 13.23 and 7.5 μM, respectively. While the free energy of binding and estimated inhibition constant for the ‘Cur-Aβ interaction’ was determined to be −3.61 kcal/mol and 136.2 µM (Table 2). Chloramphenicol (PubChem ID: 5959) was used as a negative control. The free energy of binding for the chloramphenicol with Aβ was found to be +1.16 kcal/mol. One H-bond UNK1:H43-GLY37: O was present in the ‘Cur-Aβ interaction’. The H-bonds distances were 2.13938 Å. Eight carbon atoms of Cur, namely C5, C4, C16, C10, C13, C8, C6, and C7 were found to be in hydrophobic interactions with amino acid residues Ala21, Leu34 and Val36 of the Aβ protein. C4 and C5 were found to be interacting with Ala21. C10, C13, and C16 were found to be interacting with Leu34 and C4, C5, C6, C7, and C8 were found to interact with Val36. In these interactions there are not any polar, pi-pi and cation-pi bonds present. ‘Van der Waals’, ‘hydrogen bond’ and ‘desolvation’ energy components for ‘Cur-Aβ interaction’ were −5.56 kcal/mol while the ‘electrostatic’ energy component was found to be −0.03 kcal/mol. The internal molecular energy component was found to be −5.59kcal/mol (Table 2). Similarly, on the other hand, two H-bonds ALA21:H-UNK1:O26 and UNK1:H52-PHE19:O were present in the ‘Vnc-Aβ interaction’. The H-bonds distances were 1.87477 and 2.11324 Å, respectively. Two carbon atoms of Vnc, namely C2 and C15 were found to be involved in hydrophobic interactions with amino acid residues Ala21 and the Aβ protein. One of the nitrogen atoms N1 and one H12 atom of Vnc were observed to make polar bonds with one amino acid residue Asp23, but no pi-pi and cation-pi bonds were present. ‘Van der Waals’, ‘hydrogen bond’ and ‘desolvation’ energy components for ‘Vnc-Aβ interaction’ were −5.33 kcal/mol while the ‘electrostatic’ energy component was found to be -0.78 kcal/mol. The internal molecular energy component was found to be −6.11kcal/mol (Table 2). In continuation, two H-bonds MET35:H-UNK1:O18 and UNK1:H46-GLY33:O were present in the ‘Ajm-Aβ interaction’. The H-bonds distances were 1.81314 and 1.87937 Å, respectively. Four carbon atoms of Ajm, namely C15, C18, C19, and C13 were found to be involved in hydrophobic interactions with amino acid residues Ala21, Val24 of the Aβ protein, in which C15 had interacted with Ala21; C18 and C19 were found to interact with Val24 and C13 interacted with Leu34. One of the hydrogen atoms H8 of Vnc was observed to make polar bonds with one amino acid residue Asp23. In this interaction, there were no pi-pi and cation-pi bonds. ‘Van der Waals’, ‘hydrogen bond’ and ‘desolvation’ energy components for ‘Ajm-Aβ interaction’ were −6.27 kcal/mol while the ‘electrostatic’ energy component was found to be -0.98 kcal/mol. The internal molecular energy component was found to be −7.25 kcal/mol (Table 2). Similarly, one H-bond GLY37:H-UNK1:O32 was present in the ‘Eme-Aβ interaction’. The H-bond distance was 2.41182 Å. Six carbon atoms of Eme, namely C5, C6, C7, C28, C27, C19 were found to be involved in hydrophobic interactions with amino acid residues Ala21, Leu34 and Val36 of the Aβ protein, in which C5, C6, C7 interacted with Ala21; C28, C27, C19 were found to interact with Leu34 and C5 and C6 interacted with Val36. Two hydrogen atoms H29 and H30 of Eme were observed to make polar bonds with one amino acid residue, Asp23. In this interaction, there were no pi-pi and cation-pi bonds present. ‘Van der Waals’, ‘hydrogen bond’ and ‘desolvation’ energy components for ‘Eme-Aβ interaction’ were −8.2 kcal/mol while the ‘electrostatic’ energy component was found to be −0.88 kcal/mol. The internal molecular energy component was found to be −9.08 kcal/mol (Table 2). It is appropriate to mention that ΔG value obtained through computational study can only propose the efficiency of binding for a ligand-enzyme pair [24].

### 2.1. Protein-Protein Interaction Study

An earlier study reported the suitable interface of RAGE and Aβ at the luminal membrane of the BBB, proposing that RAGE acts as a transporter protein for circulating Aβ across the BBB [25]. This fact is supported in another research where RAGE mediates the entry of Aβ1-40 and Aβ1-42 into the hippocampus and cortex across the BBB [26]. The Aβ-RAGE connection at the BBB does not just result in neurovascular stress and articulation of proinflammatory cytokines (TNF-𝛼 and IL-6), in addition, it prompts diminished cerebral bloodstream by improving the emission of endothelin-1 to instigate vasoconstriction [27].

We therefore further deciphered the interaction impact of Aβ on RAGE in terms of binding efficiency and interacting amino acid residues. For this reason, we connected the Z-dock technique by figuring out the Z-dock score of protein-protein association between Aβ and RAGE and compared it with docked protein complexes of (Aβ+Ajm, Aβ+Eme, and Aβ+Vnc) with RAGE. The Z-dock score for Aβ and RAGE interaction was found to be 1269.55. Met35, Asp23, Ser26, Asn27 and Lys28 of the Aβ and Trp61, Arg114, Arg116, Val117 amino acid residues of RAGE were found to participate in the interaction (Figure 2).

In the complex of Aβ and RAGE, seven H-bonds were found in which Met35 interacts with Trp61. Two amino acid residues of Asp23 were found to interact with Arg114, Ser26 interacted with Arg116, Asn27 interacted with Arg116, and two amino acid residues of Lys28 interacted with two amino acid residues of Val117 (Table 3). 

The Z-dock score of the complexes (Aβ+Ajm, Eme+Aβ, and Aβ+Vnc) with RAGE was found to be 911.83, 940.69 and 907.98, respectively. In these complexes, no H-bond formation had taken place. The interaction complexes are shown in (Figure 3).

Results clearly reflected a significant decrease in the Z-dock score of Aβ and RAGE from 1269.55 to 911.83, 940.69 and 907.98 for (Aβ+Ajm, Eme+Aβ, and Aβ+Vnc) with RAGE (Table 4) along with loss of H-Bond formation.

With the help of the above results, one could say that the ligands Ajm, Eme and Vnc play a very important role in avoiding the interaction between Aβ and RAGE.

### 2.2. Comparison of Vnc, Ajm, and Eme With Positive Control Curcumin

Curcumin has a BBB of 0.913545 which is higher than that of Vnc (0.871568) and Eme (0.875498), but lower than Ajm (1.9897). Human Intestinal Absorption (HIA) value for curcumin was 94.40% (Appendix A) which is higher than that of Ajm (93.31%), but lower than Vnc (95.95%) and Eme (96.59%). Thus, Vnc and Eme may be better absorbed by the human body than curcumin. With the help of the PreADME server, we have analyzed the toxicity of Vnc, Ajm, Eme leads and curcumin. Curcumin, Vnc, and Eme are nonmutagenic while Ajm is mutagenic. Vnc, Ajm, and Eme are non-carcinogenic in the mouse-like curcumin. Curcumin and Eme are carcinogenic in rats while Vnc and Ajm are non-carcinogenic in rats. 

Thereby, the values of Ki and ΔG, interacting amino acid residues, H-bond, polar and hydrophobic interactions and properties of Vnc, Ajm and Eme against curcumin obtained through computational studies probably suggest the efficacy of the proposed leads in AD therapy. The “computational” studies, with reference to the protein (Aβ) and ligands (Vnc, Ajm, and Eme) and protein-protein interactions, are expected to frame the premise of future treatment against a few neurological disarranges [28]. This article may be considered as an extension of our ongoing research work whereby we have reported inhibition of the related neuro-toxic Aβ by selected ligands and an inhibition study with the well-known ligand, curcumin.

## 3. Materials and Methods

### 3.1. Preparation of Receptor-Protein Structures

The three-dimensional structures of the β-amyloid (Aβ) (PDB ID: 2BEG) and RAGE (PDB ID: 2ENS) were obtained using RCSB-PDB (www.rcsb.org) (Figure 4). The PDB file was cleaned and the heteroatoms (HETATM) of the receptor were removed manually since these are non-standard residues of the protein [29]. 

Chimera was used for energy minimization. The steric collision was detached for the steepest gradient minimization. The steepest descent steps and the size were 1000 and 0.02 Å, respectively. The consumer gradient steps were 1000 and the gradient step size was 0.02 Å [30,31].

### 3.2. Preparation of Ligand Structure

The simplified molecular input line entry specification notations of the inhibitors ajmalicine (Ajm), emetine (Eme) and vincamine (Vnc) were obtained from the PubChem database. The online display of CORINA (http://www.molecular-networks.com/ products/ Corina) was used to build the 2D structures of the ligands (Figure 5A–D). Before docking, these inhibitors were energy minimized by Chimera software [32] and saved in PDP format. Gasteiger charges [33] were applied to the ligands and they were further exposed to single-step minimization.

### 3.3. Molecular Interaction Study

Molecular interaction examinations were performed via the Autodock version 4.2 suite with the Cygwin interface instrument [34,35]. Molecular docking methods were used for target and ligand interaction to obtain the top conformations on the foundation of binding energy (kcal/mol). Before applying the docking algorithm, we marked all water molecules in proteins that had been expelled from the focused protein structure. The hydrogen molecules were included in the target molecules. After, Kollman united charges and solvation parameters were applied to proteins. Gasteiger partial charges were added to the ligands atom. A framework box was set to cover the greatest piece of chosen protein for ligand cooperation. The value was set to standard 60 × 60 × 60 Å in X, Y and Z as the organization of the network point with the default estimation of framework focuses separating at 0.375 Å. The Lamarckian genetic algorithm (LGA) was connected at the receptor protein and ligand for adaptable docking calculations [36,37]. The LGA parameters, as well as the populace measure, vitality assessments, transformation rate, hybrid rate, and step size, were set to: 150; 2,500,000; 27,000; 0.02; 0.8 and; 0.2 Å, respectively. The LGA runs were set to a standard of 10 runs. We watched all adaptations of protein with ligand complex. They were then examined for the connection orientations in terms of binding energies of the docked structure utilizing Discovery Studio Visualizer.

### 3.4. Protein-Protein Interaction Analysis

Z-dock was utilized for protein-protein interactions (Aβ+RAGE) [38] as well as for protein-ligand complexes and protein (Aβ+Ajm, Aβ+Eme, and Aβ+Vnc & RAGE) interactions. Z-dock is one of the most unbeaten suites that encompass large calculation facility in Critical Assessment of Predicted Interactions (CAPRI) [39]. Z-dock is an original phase rigid body molecular docking algorithm that utilizes a fast Fourier transform (FFT) calculation to calculate progress performance for translational searching [40]. 

## 4. Conclusions

The present study proposes that all the selected compounds can be absorbed by the human body, by passing through the BBB, and have high inhibition potential. With the help of docking, we have proposed the inhibition of Aβ and degradation of amyloid peptide aggregation. Our compounds have shown the higher efficiency to bind with Aβ (Vnc: −5.45 Kcal/Mol, Ajm: −6.66 Kcal/Mol, Eme: −6.99 Kcal/Mol) compared to standard Cur: −3.61 Kcal/Mol. Z-dock scores calculated from protein-protein and complex-protein connections further support that the selected compounds have potential for disaggregation. 

## Figures and Tables

**Figure 1 molecules-24-03233-f001:**
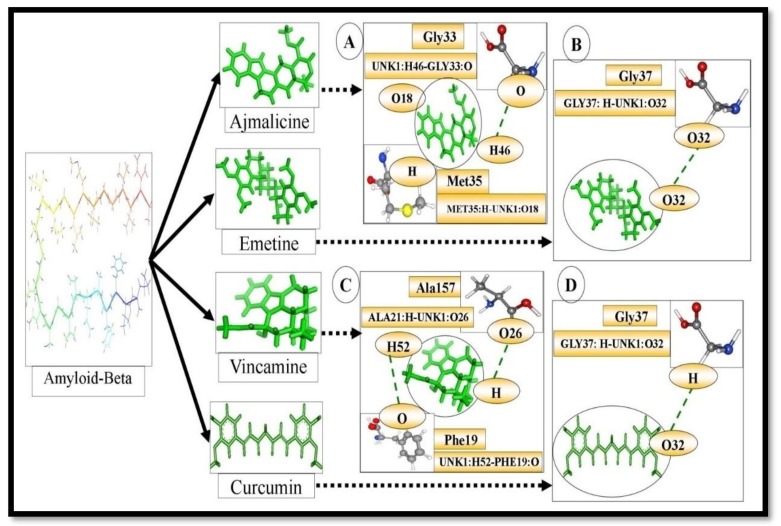
(**A**) The complex shows interacting amino acid residues and hydrogen bonds formed between compound ajmalicine (the ligand, ajmalicine, has been shown in green ‘stick’ representation) and β-Amyloid (**B**) The complex interacting amino acid residues and hydrogen bonds formed between compound emetine (the ligand, emetine, has been shown in green ‘stick’ representation) and β-Amyloid (**C**) The complex interacting amino acid residues and hydrogen bonds formed between compound vincamine (the ligand, vincamine, has been shown in green ‘stick’ representation) and β-Amyloid. (**D**) The complex interacting amino acid residues and hydrogen bonds formed between compound curcumin (the ligand, curcumin, has been shown in green ‘stick’ representation) and β-Amyloid.

**Figure 2 molecules-24-03233-f002:**
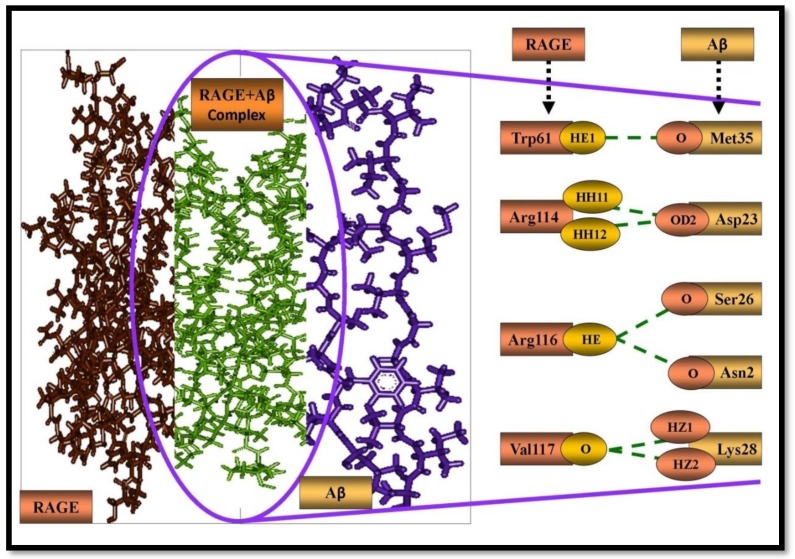
Protein-protein interaction. (**A**) Aβ, (**B**) RAGE, (**C**) the complex of Aβ and RAGE obtained by the protein-protein docking method. Purple and brown stick color representations are the amino acid residues of Aβ and RAGE, respectively, involved in H-bond formation.

**Figure 3 molecules-24-03233-f003:**
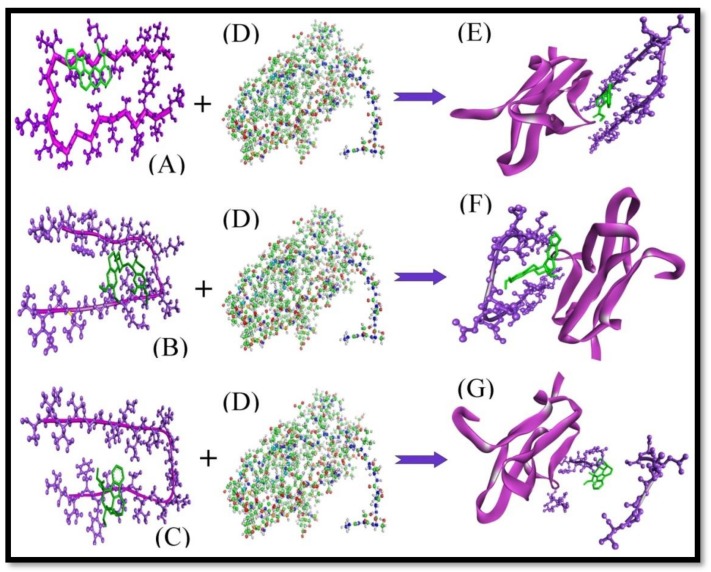
The complex-protein interaction. (**A**) The complex of Aβ+Ajm, (**B**) the complex of Aβ+Eme, (**C**) the complex of Aβ+Vnc. (**D**) Structure of RAGE protein. (E) The interacting complex structure of (Aβ+Ajm) with RAGE obtained by the protein-protein docking method. (**F**) The interacting complex structure of (Aβ+Eme) with RAGE obtained by the protein-protein docking method. (**G**) The interacting complex structure of (Aβ+Vnc) with RAGE obtained by the protein-protein docking method.

**Figure 4 molecules-24-03233-f004:**
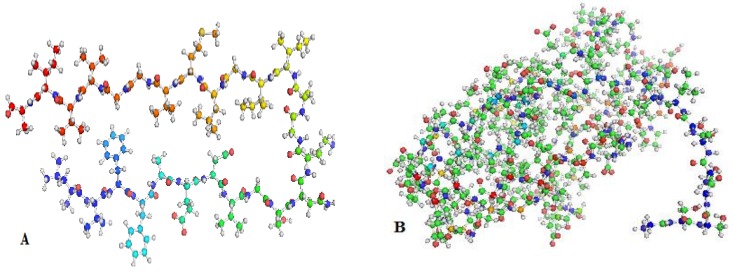
The three dimensional structures of the protein. (**A**) β-Amyloid (PDB ID: 2BEG). (**B**) Receptor for advanced glycation end products (RAGE) (PDB ID: 2ENS).

**Figure 5 molecules-24-03233-f005:**
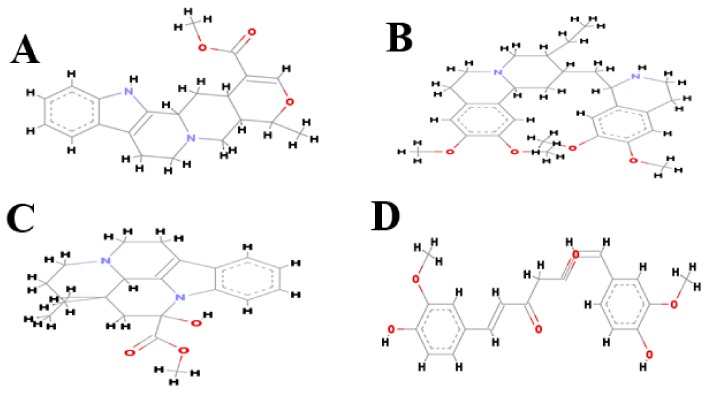
2D chemical structure of the compounds: (**A**) ajmalicine, (**B**) emetine, (**C**) vincamine, (**D**) curcumin.

**Table 1 molecules-24-03233-t001:** Interacting amino acid residues and the H-bond distance between β-Amyloid and natural compound.

S.N	Target	Ligands Name	Interaction Amino Acid	H-bond Distance (Å)	H-bond
**1.**	Β-Amyloid-	Vincamine	Phe19, Phe20, Ala21, Asp23, Ile32, Gly33, Leu34, Met35, and Val36	1.87477 2.11324	ALA21:H-UNK1: O26 UNK1: H52-PHE19:O
**2.**		Ajmalicine	Phe19, Phe20, Ala21, Gly22, Asp23, Ile32, Gly33,Leu34, Met35, and Val36	1.81314 1.87937	MET35:H-UNK1: O18 UNK1: H46-GLY33:O
**3.**		Emetine	Ala21, Gl22, Asp23, Gly33, Leu34, Met35, Val36, and Gly37	2.41182	GLY37:H-UNK1: O32
**4.**		Curcumin	Ala21, Glu22, Asp23, Val24, Gly25, Leu34, Met35, Val36, and Gly37.	2.13938	UNK1: H43-GLY37:O

**Table 2 molecules-24-03233-t002:** Interaction energies of Aβ with ligands (Kcal/mol) obtained from molecular docking analysis.

S. No	Target	Ligand Name	Binding Energy (Kcal/mol)	Inhibition Constant (Ki)	vdw+hb+ Desolvation Energy (Kcal/mol)	Internal Molecular Energy (Kcal/mol)	Electrostatic Energy (Kcal/mol)
**1.**	β-Amyloid	Vincamine	−5.45	249.96 µM	−5.33	−6.11	−0.78
**2.**		Ajmalicine	−6.66	13.23 µM	−6.27	−7.25	−0.98
**3.**		Emetine	−6.99	7.5 µM	−8.2	−9.08	−0.88
**4.**		Curcumin	−3.61	136.2 µM	−5.56	−5.59	−0.03

**Table 3 molecules-24-03233-t003:** Acid residues involved in hydrogen bond formation in the protein-protein interaction.

S S.No	Amino Acid Residues of Aβ	Amino Acid Residues of RAGE	Donor Atom	Acceptor Atom	H-Bonds	Distance of H-Bond (Å)
**1.**	Met35	Trp61	TRP61:HE1	MET35:O	TRP61:HE1 - MET35:O	2.31028
**2.**	Asp23	Arg114	ARG114:HH11	ASP23:OD2	ARG114:HH11 -ASP23:OD2	1.40911
**3.**	Asp23	Arg114	ARG114:HH12	ASP23:OD2	ARG114:HH12 -ASP23:OD2	2.02749
**4.**	Ser26	Arg116	ARG116:HE	SER26:O	ARG116:HE - SER26:O	2.48357
**5.**	Asn27	Arg116	ARG116: HE	ASN27:O	ARG116: HE - ASN27:O	2.04783
**6.**	Lys28	Val117	LYS28:HZ1	VAL117:O	LYS28:HZ1 - VAL117:O	2.15927
**7.**	Lys28	Val117	LYS28:HZ2	VAL117:O	LYS28:HZ2 - AVAL117:O	1.10912

**Table 4 molecules-24-03233-t004:** Score of protein-protein and complex-protein interactions.

S.No.	Protein-Protein interaction	Z-dock Score
**1.**	Aβ+RAGE	1269.55
	**Complex-Protein Interaction**	**Z-dock Score**
**2.**	(Aβ+Ajm)+ RAGE	911.83
**3.**	(Aβ+Eme)+ RAGE	940.69
**4.**	(Aβ+Vnc)+ RAGE	907.98

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
