# Peer review of "Computational Study of Natural Compounds for the Clearance of Amyloid-Βeta: A Potential Therapeutic Management Strategy for Alzheimer’s Disease"

_molecules, 2019, doi:10.3390/molecules24183233_

Round 1

Reviewer 1 Report

The authors are here reporting a manuscript entitled “Screening of Natural
Compounds for the Clearance of Amyloid-Βeta: A Potential Therapeutic Management Strategy for Alzheimer’s disease”.

I reported in my previous revision "The overall idea could be of interest, although in absence of experimental proofs that sustain the predicted results the work is of limited impact and significance".

I apologize if this sentence has created misunderstanding to the authors, I meant that the overall "idea" and not the work by itself could be of interest, as mentioned by the authors in the comment's response. It may be interesting to evaluate these compounds in models of neurological disorders, but the work itself, in absence of experimental proofs that sustain the predicted results, is of limited impact and significance.

At the state of the art, I recommend rejection.

Author Response

"The authors are here reporting a manuscript entitled “Screening of Natural Compounds for the Clearance of Amyloid-Βeta: A Potential Therapeutic Management Strategy for Alzheimer’s disease”.

Comment 1: I reported in my previous revision "The overall idea could be of interest, although in absence of experimental proofs that sustain the predicted results the work is of limited impact and significance".

I apologize if this sentence has created misunderstanding to the authors, I meant that the overall "idea" and not the work by itself could be of interest, as mentioned by the authors in the comment's response. It may be interesting to evaluate these compounds in models of neurological disorders, but the work itself, in absence of experimental proofs that sustain the predicted results, is of limited impact and significance.

Response 1: Thank you for appreciating the idea proposed our work. Indeed, this work will provide the leads to the researchers in the field of neurobiology. There are very few researchers working in this field due to very limited leads and restrictions of the experiments. We have explained everything in a very systematic manner. We have designed this study to propose the new leads. This pre clinical study could be helpful to the researcher for the development of new leads for better treatment. It is highly impact work on the basis of computation work which is most valuable for pre clinical work as it provide the suitable base and target point for the wet lab study.

Reviewer 2 Report

In the present study, “Screening of natural compounds for the clearance of amyloid-beta: A potential therapeutic management strategy for Alzheimer’s disease”, Ahmad and colleagues performed an in silico prediction of test compounds-amyloid-beta interaction along with the RAGE and concluded that the test compounds are capable of inhibiting Aβ-RAGE interaction. In silico prediction of drug-target interaction has great importance in drug discovery. The study is well-designed. However, looking at the title, readers expect at least in vitro results, which is lacking. Moreover, the number of compounds studied is very less. Authors could have performed in vitro amyloid disaggregation assay before the in silico prediction or at least compared the interactions of ligands within different amyloid fragments. I suggest authors revise the title and/or perform some additional work to enhance the paper quality.

Additionally, please revise the followings.

In abstract (line 24-26): In current study we have hereby predicted the interaction potential of …… with the enzymes concerned in the treatment of AD against the standard control curcumin to validate the enzyme-ligand results. However, there is no report on enzyme-ligand interaction. This study solely predicts interactions of compounds and Aβ and RAGE.

Please re-write the compounds name in sentence case in the abstract. Once acronyms are assigned for compounds, use the acronyms uniformly.

In section 3, the same sentence-structures have been used repeatedly to present the binding results of the test and reference compounds with amyloid-beta. Readers feel monotonous in such case. So authors should revise this section. Also, the section lacks the discussion part. As the section indicates ‘Results and Discussion’, the results should be discussed appropriately. How do you justify the higher binding energy of curcumin compared to the test ligands? What about the stability of the ligand-Aβ complex?

In line 150-151, does it mean eight carbon atoms of curcumin involved ten hydrophobic interactions? If so, please correct that sentence and also in line 175-177.

Author Response

Response of Comments of Reviewer #2

Comments and Suggestions for Authors

In the present study, “Screening of natural compounds for the clearance of amyloid-beta: A potential therapeutic management strategy for Alzheimer’s disease”, Ahmad and colleagues performed an in silico prediction of test compounds-amyloid-beta interaction along with the RAGE and concluded that the test compounds are capable of inhibiting Aβ-RAGE interaction. In silico prediction of drug-target interaction has great importance in drug discovery. The study is well-designed. However, looking at the title, readers expect at least in vitro results, which is lacking. Moreover, the number of compounds studied is very less. Authors could have performed in vitro amyloid disaggregation assay before the in silico prediction or at least compared the interactions of ligands within different amyloid fragments.

Comment 1: I suggest authors revise the title and/or perform some additional work to enhance the paper quality.

Response 1: Authors are thankful to the esteemed reviewer for constructive feedback. The title of this manuscript has been revised.

______________________________________________________________________________

Additionally, please revise the followings.

Comment 2: In abstract (line 24-26): In current study we have hereby predicted the interaction potential of …… with the enzymes concerned in the treatment of AD against the standard control curcumin to validate the enzyme-ligand results. However, there is no report on enzyme-ligand interaction. This study solely predicts interactions of compounds and Aβ and RAGE.

Response 2: The line number 24-26 has been revised according to the suggestion of esteemed reviewer in this manuscript.

Comment 3: Please re-write the compounds name in sentence case in the abstract. Once acronyms are assigned for compounds, use the acronyms uniformly.

Response 3: The names of compounds have been revised according to the suggestion in this manuscript.

______________________________________________________________________________

Comment 4: In section 3, the same sentence-structures have been used repeatedly to present the binding results of the test and reference compounds with amyloid-beta. Readers feel monotonous in such case. So authors should revise this section. Also, the section lacks the discussion part. As the section indicates ‘Results and Discussion’, the results should be discussed appropriately. How do you justify the higher binding energy of curcumin compared to the test ligands? What about the stability of the ligand-Aβ complex?

Response 4: In section 3, the same sentence-structures have been revised in this manuscript. The binding energy is the change in energy of the receptor and ligand due to complex formation and it is due to specific interactions between the ligand and the receptor and the contribution due to changes in movement. The successful integration of binding energy with the usual virtual screening lowest energy score can improve the physiological relevance of well established approaches in virtual screening for rational lead discovery. In virtual screening experiments the binding energy landscape analysis should be applied after all clearly structurally incompatible chemicals are filtered based on the lowest energy score. After this step all binding conformations within the boundaries of the native binding phase have to be investigated and analyzed in detail.  Lower the binding energy more stable is the complex. 

Reference:

Pantsar T, Poso A Binding Affinity via Docking: Fact and Fiction. Molecules. 2018 Jul 30; 23(8). pii: E1899. doi: 10.3390/molecules23081899. Grigoryan AV, Wang H, Cardozo TJ. Can the energy gap in the protein-ligand binding energy landscape be used as a descriptor in virtual ligand screening? PLoS One. 2012; 7(10):e46532. doi: 10.1371/journal.pone.0046532.  

_____________________________________________________________________________

Comment 5: In line 150-151, does it mean eight carbon atoms of curcumin involved ten hydrophobic interactions? If so, please correct that sentence and also in line 175-177.

Response 5: The line no. 150-151 and 175-177 has been revised in this manuscript

____________________________________________________________________________

Reviewer 3 Report

Manuscript ID: molecules-576767: Screening of Natural Compounds for the Clearance of Amyloid-Βeta: A Potential Therapeutic Management Strategy for Alzheimer’s Disease This paper describes the in silico analysis of the interactions among Abeta, RAGE, and natural compounds. Although these results need be re-evaluated in vivo (cultured cells or animals), the promising results may help for the development of the drugs of Alzheimer's disease (AD). This report is worth being published in the Journal as a starting report. The followings should be addressed to help increase the value of this report. Major Points
(1) Negative Control for the interaction analysis.
It was very nice to use Curcumin as a positive control. To more reliably show the binding, the data of a negative control should be added. Because Abeta may have a broad binding spectrum, the selection of a negative control may be difficult. One suggestion is Minocycline, Erythromycin, or Chloramphenicol, since their structures are as complex as those of Ajmalicine, Emetine, and Vincamine. In addition, these antibiotics have no curing effects on AD.
(2) Discussion of the feasibility of the described interactions in vivo
The molecules are dissolved in very complex solution in vivo. The concentration of proteins, lipids, or other organic or inorganic solute molecules is high. These conditions should affect the interaction of Abeta, RAGE, and the natural compounds described.
Minor Points
(a) Abeta40 (containing 40 amino-acid residues) has properties different from Abeta42. The possibility of whether these differences might have some effects on the present results should be briefly discussed
(b) Line 25 to 26 " of the natural compounds Vincamine, Ajmalicine and Emetine with the *enzymes* concerned in the treatment of AD". It is unclear which enzymes are referred. Are they secretases? (c) Line 243 to 244 "This article may be considered as an extension of our ongoing research work whereby we have reported*" The reference of * is necessary. (d) The full spellings of BBB (line 21) and RAGE (line 20 and line 44) are necessary.  

Author Response

Response of Comments of Reviewer #3

Manuscript ID: molecules-576767: Screening of Natural Compounds for the Clearance of Amyloid-Βeta: A Potential Therapeutic Management Strategy for Alzheimer’s Disease. This paper describes the in silico analysis of the interactions among Abeta, RAGE, and natural compounds. Although these results need be re-evaluated in vivo (cultured cells or animals), the promising results may help for the development of the drugs of Alzheimer's disease (AD). This report is worth being published in the Journal as a starting report. The followings should be addressed to help increase the value of this report. 

Major Points
Comment (1): Negative Control for the interaction analysis. It was very nice to use Curcumin as a positive control. To more reliably show the binding, the data of a negative control should be added. Because Abeta may have a broad binding spectrum, the selection of a negative control may be difficult. One suggestion is Minocycline, Erythromycin, or Chloramphenicol, since their structures are as complex as those of Ajmalicine, Emetine, and Vincamine. In addition, these antibiotics have no curing effects on AD. 

Response 1: Authors are thankful to the reviewer#3 for constructive suggestion. According to the suggestion of reviewer, Chloramphenicol has been tested against Abeta. The free energy of binding was found to be +1.16kcal/mol, which is showing least amount of energy and lead to non bonding interaction between “Abeta and Chloramphenicol” complex. So it can be used as a negative control for Abeta.

_____________________________________________________________________________

Comment (2): Discussion of the feasibility of the described interactions in vivo.
The molecules are dissolved in very complex solution in vivo. The concentration of proteins, lipids, or other organic or inorganic solute molecules is high. These conditions should affect the interaction of Abeta, RAGE, and the natural compounds described. 

Response 2: The selected compounds have ability to destroy the structure of Abeta, it is shown by interaction structure. The inhibition of Abeta is useful for the AD treatment. And selected compound having stability like a drug molecule to cure the disease. The solubility of the compounds are listed below:

Vincamine: Melting Point- 5°C; Solubility- In water, 62 mg/L Ajmalicine: Melting Point- 5 to 263 °C; 33.3 mg soluble in 1 mL of chloroform Emetine: Melting Point- 0°C; Solubility: moderately sol in dilute ammonium hydroxide; sol in ether, alcohol, acetone; slightly sol in chloroform

_____________________________________________________________________________

Minor Points

 Comment (a): Abeta40 (containing 40 amino-acid residues) has properties different from Abeta42. The possibility of whether these differences might have some effects on the present results should be briefly discussed 

Response (a): Abeta42 is a main peptide which forms the plaque in the brain, finally cause the Alzheimer’s disease (AD). So, in this study authors target the Abeta42 for the better treatment of AD.

____________________________________________________________________________

Comment (b): Line 25 to 26 " of the natural compounds Vincamine, Ajmalicine and Emetine with the *enzymes* concerned in the treatment of AD". It is unclear which enzymes are referred. Are they secretases?

Response (b): The Line 25 to 26 has been revised. It is Aβ peptide which is a target used in the interaction of compounds Vincamine, Ajmalicine and Emetine.

_____________________________________________________________________________

Comment (c): Line 243 to 244 "This article may be considered as an extension of our ongoing research work whereby we have reported*" The reference of * is necessary. 

Response (c): Reference is incorporated in this manuscript according to the suggestion.

____________________________________________________________________________

Comment (d): The full spellings of BBB (line 21) and RAGE (line 20 and line 44) are necessary

Response (d): The full spellings of BBB (line 21) and RAGE (line 20 and line 44) have been revised according to the suggestion of esteemed reviewer.

Prof. Dr. Haroon Khan 
==============================

Pharm., M. Phil., Ph. D.,

Chairman Department of Pharmacy,

Faculty of Chemical & Life Sciences 

Abdul Wali Khan University Mardan. 

Regional Editor- Endocrine, Metabolic & Immune Disorders - Drug Targets

Editorial Board Member: Food and Chemical Toxicology  

Editorial Board Member: Metabolic Brain Disease

Editorial Board Member: Phytomedicine

Member of INPST

Honorary Faculty Members of NGCEF

Round 2

Reviewer 1 Report

The authors have replied to my comments by using the same response they have used for the second revision.
As I mentioned, the work is not significant and this should not be published in a journal with high impact.

This manuscript is a resubmission of an earlier submission. The following is a list of the peer review reports and author responses from that submission.

Round 1

Reviewer 1 Report

Revision molecules-558844:

The authors are here reporting a manuscript entitled “Screening of Natural Compounds for the Clearance of Amyloid-Βeta: A Potential Therapeutic Management Strategy for Alzheimer’s Disease”. The overall idea could be of interest, although in absence of experimental proofs that sustain the predicted results the work is of limited impact and significance. At the state of the art I suggest a rejection.

Reviewer 2 Report

This manuscript describes the in silico prediction of the interaction between three natural products (compounds Vincamine, Ajmalicine and Emetine) with two enzymes concerned in the treatment of Alzheimer disease: A? and RAGE, and comparing the results against the standard control Curcumin. The manuscript fit within the scopes of Molecules. However there are several points that the authors should address before to be published:

·         Curcumin is considered one of the worst offenders of the PAINS (pan-assay interference compounds) family of compounds. These are promiscuous whose optimizations have been fraught with challenges. Curcumin can produce a covalent and irreversible light-induced reaction with proteins, as well as chelation with protein active metal sites. Authors should explain the participation of this natural product in this current work and a full discussion on how to avoid PAINS.

·         Authors must improve the quality of chemical structures of figure 2 and also include curcumin structure.

·         Authors must validate the docking procedure by re-docking the crystal ligand  (if it´s available) for Amyloid-Beta (PDB ID: 2BEG), and mention how well the bioactive conformation was reproduced by adding the RMSD value. Do the same for RAGE (PDB code: 2ENS)

·         2-D interaction maps could be welcomed in order to visualize the main interactions of the four natural products and its corresponding docked proteins.

Through a careful reading of the manuscript, I found this paper acceptable for publication, pending these minor corrections.

Reviewer 3 Report

The authors predicted the interaction potential of the natural compounds Vincamine, Ajmalicine and Emetine with the enzymes concerned in the treatment of AD against the standard control Curcumin. Protein-protein interaction studies have also been carried out to see their potential to inhibit the binding process of A? and RAGE.

·    I do not feel as if this manuscript completely complies with the scope of this journal.

·    This work lacks originality and novelty; it seems an extension of the previous papers.

·   Abstract does not provide a concise and complete summary.

·  Only three compounds have been studied and the authors do not explain why only these three compounds were selected.

·   The authors do not discuss the docking results based on the structure of the ligand and they only make a compendium of interactions with the receptors.

·     Figure 2 should be changed. Compounds must be drawn more carefully as the cycles are deformed.

·     Figure 3 should be changed in order to be clearer.